# Smoking and quit attempts during pregnancy and postpartum: a longitudinal UK cohort

Sue Cooper,[1] Sophie Orton,[1] Jo Leonardi-Bee,[2] Emma Brotherton,[1] Laura Vanderbloemen,[3] Katharine Bowker,[1] Felix Naughton,[4] Michael Ussher,[5] Kate E Pickett,[6] Stephen Sutton,[7] Tim Coleman[1]

[1]Division of Primary Care, University of Nottingham, University Park, Nottingham, UK
[2]Division of Epidemiology and Public Health, University of Nottingham, Nottingham, UK
[3]Department of Primary Care and Public Health, Imperial College London, London, UK
[4]School of Health Sciences, University of East Anglia, Norwich, Norfolk, UK
[5]Population Health Research Institute, St George's University of London, London, UK
[6]Department of Health Sciences, University of York, York, UK
[7]Behavioural Science Group, University of Cambridge, Cambridge, UK

**Correspondence to**
Dr Sue Cooper;
Sue.cooper@nottingham.ac.uk

## ABSTRACT

**Objectives** Pregnancy motivates women to try stopping smoking, but little is known about timing of their quit attempts and how quitting intentions change during pregnancy and postpartum. Using longitudinal data, this study aimed to document women's smoking and quitting behaviour throughout pregnancy and after delivery.

**Design** Longitudinal cohort survey with questionnaires at baseline (8–26 weeks' gestation), late pregnancy (34–36 weeks) and 3 months after delivery.

**Setting** Two maternity hospitals in one National Health Service hospital trust, Nottingham, England.

**Participants** 850 pregnant women, aged 16 years or over, who were current smokers or had smoked in the 3 months before pregnancy, were recruited between August 2011 and August 2012.

**Outcome measures** Self-reported smoking behaviour, quit attempts and quitting intentions.

**Results** Smoking rates, adjusting for non-response at follow-up, were 57.4% (95% CI 54.1 to 60.7) at baseline, 59.1% (95% CI 54.9 to 63.4) in late pregnancy and 67.1% (95% CI 62.7 to 71.5) 3 months postpartum. At baseline, 272 of 488 current smokers had tried to quit since becoming pregnant (55.7%, 95% CI 51.3 to 60.1); 51.3% (95% CI 44.7 to 58.0) tried quitting between baseline and late pregnancy and 27.4% (95% CI 21.7 to 33.2) after childbirth. The percentage who intended to quit within the next month fell as pregnancy progressed, from 40.4% (95% CI 36.1 to 44.8) at baseline to 29.7% (95% CI 23.8 to 35.6) in late pregnancy and 14.2% (95% CI 10.0 to 18.3) postpartum. Postpartum relapse was lower among women who quit in the 3 months before pregnancy (17.8%, 95% CI 6.1 to 29.4) than those who stopped between baseline and late pregnancy (42.9%, 95% CI 24.6 to 61.3).

**Conclusions** Many pregnant smokers make quit attempts throughout pregnancy and postpartum, but intention to quit decreases over time; there is no evidence that smoking rates fall during gestation.

## INTRODUCTION

Smoking in pregnancy is associated with increased risks of miscarriage, stillbirth, prematurity, low birth weight, perinatal morbidity and mortality, neonatal and sudden infant death, infant respiratory

### Strengths and limitations of this study

► As far as we are aware, this is the only study to investigate timing of quit attempts and propensity to stop smoking during pregnancy and postpartum and to quantify longitudinal changes.
► Smoking behaviour is self-reported rather than validated; misreporting due to recall bias may have been minimised by collecting data at three time points and by there being no expectation that they should try to stop smoking.
► Later survey findings were adjusted using multiple imputation to help address non-response bias due to attrition.
► As the study was conducted in just one geographical area of the UK and participants were predominantly white British, findings might not be generalisable; however, the demographic profile of participants was similar to that of other UK cohorts of pregnant smokers.

problems, poorer infant cognition and adverse infant behavioural outcomes.[1 2] Internationally, large numbers of pregnant women smoke, with rates between 12% and 22% in high-income countries[3–6] and rates increasing in emerging and developing economies.[7] Pregnancy is probably the event which most motivates female smokers to try quitting; for example, in the UK, over 50% of pregnant smokers try to stop[5] and pregnant women are, therefore, particularly likely to be interested in receiving cessation support. Some health systems systematically offer such support; in the UK, this is largely done in early pregnancy,[8] although official guidance recommends that support is provided throughout gestation.[9]

We are not aware of any studies that have investigated when, in pregnancy, smokers have the greatest propensity to try stopping, the timing of any quit attempts and potential influences on this. Outside of pregnancy and postpartum, most adults tend to have fairly

stable smoking behaviour.[10] Although overall smoking rates in pregnancy have declined, a significant proportion of women continue to smoke throughout pregnancy.[11] However, many women who smoke before pregnancy have varied smoking behaviour after conception,[5 11–16] and although it is logical to try to minimise fetal exposure to tobacco smoke by offering cessation support in early pregnancy, support may be welcomed at other times in gestation. In addition, of those that do stop, many relapse within the first few months postpartum.[17 18] Relatively few studies of prenatal smoking behaviour have been longitudinal,[12 13 15 16 19–23] with only two of these following up women postpartum,[12 13] and the only two studies to have been conducted in the UK are now over 20 years old.[12 19] Importantly, none of these studies asked about number of quit attempts or reported when in pregnancy women have tried to quit. To help focus smoking cessation interventions at the most effective leverage points, we need contemporary, longitudinal data on the smoking and quitting behaviours of pregnant women. Consequently, we investigate the frequency and timing of pregnant smokers' quit attempts and the factors associated with these. We also attempt to quantify individual-level changes in smoking behaviour during these times.

## METHODS

### Study design and configuration

A longitudinal cohort study was undertaken; eligible women were aged 16 years or over, 8–26 weeks pregnant and currently smoking or had smoked regularly during the 3 months immediately prior to finding out they were pregnant. Surveys were conducted at recruitment (8–26 weeks' gestation), in late pregnancy (34–36 weeks' gestation) and 3 months after delivery. Full methods and characteristics of the recruited participants, including factors associated with being a current smoker, are detailed elsewhere.[24] STROBE guidance was used for reporting.[25] Ethical approval was given by Derbyshire Research Ethics Proportionate Review Sub-Committee.

### Study setting and regimen

Women were recruited between August 2011 and August 2012 when attending routine hospital or ultrasound appointments at two antenatal clinics within Nottingham University Hospitals National Health Service Trust (City Hospital and Queen's Medical Centre). Women attending clinics first completed a screening questionnaire and those eligible and willing completed a baseline questionnaire and were sent follow-up questionnaires by post or via an email web link.

### Measurements

At baseline, 'recent ex-smokers' were women who reported not smoking currently but had done so during the 3 months before finding out they were pregnant. On later questionnaires, women who reported quitting smoking since the previous survey were also defined as 'recent ex-smokers'. At any time, if women reported smoking either every day or occasionally (smoke, but not every day), they were classified as smokers and were asked further details about this, including if they had made any quit attempts and how many of these attempts had lasted at least 24 hours. On all questionnaires, women were asked about timescales of future intentions to quit (within next 2 weeks/next 30 days/next 3 months/not planning to quit) (since finding out they were pregnant/since completing the first questionnaire/since the birth of their baby) and about urges to smoke (6-point Likert scale ranging from 'no urges' to 'extremely strong'). The Heaviness of Smoking Index (HSI) was calculated as the sum of scores from two items of the Fagerström Test of Cigarette Dependence[26] (scores range from 0 to 6; higher score indicates greater cigarette dependence). The questionnaires are available as online supplementary appendix to this article.

### Statistical analysis

To quantify the proportion of quit attempts made after the first trimester of pregnancy, we aimed to recruit 850 participants.[24] Analyses were conducted using Stata V.14.0 (StataCorp).

Descriptive statistics summarised participants' characteristics and smoking behaviour at each time point; we compared those responding to all three questionnaires with those who did not by using $X^2$ and t tests for categorical and continuous variables, respectively, with P values of <0.05 deemed significant. Characteristics found to be significantly associated with non-completion of later questionnaires, and hence with absence of smoking data were used with multiple imputation to adjust for attrition of smoking behaviour at later time points.

An exploratory analysis was performed to investigate the factors associated with reporting having made a quit attempt of any duration on the baseline questionnaire. For this analysis, items were dichotomised; six self-efficacy items that had high internal consistency (Cronbach's α=0.95)[27] were combined into a single score out of 30 with women scoring ≥25 considered to have high self-efficacy. Univariable logistic regression analysis was used to calculate an OR with 95% CIs for each variable (age,[28 29] ethnicity,[30] qualifications held,[28] previous pregnancy,[28 29 31 32] number of cigarettes smoked per day, HSI,[29 31 33] partner smoking status,[29] occupation,[34] planned or surprise pregnancy,[28 35–37] depression, long-term disability or mental illness,[38] smoking beliefs and self-efficacy[39]). Variables which showed a significant association (P<0.05) in the univariable analysis were included in a multivariable logistic regression model. Variables that achieved significance (P<0.05) remained in the multivariable model and all non-significant variables identified from the univariable analysis were re-entered into the model consecutively to assess whether they became significant. The final multivariable model included only significant variables (P<0.05). A likelihood ratio test identified that age should be included in the multivariable analysis

as a continuous variable. Where collinearity between variables was anticipated (eg, the number of cigarettes smoked per day and HSI), we included the variable that resulted in a better fitting model. As this analysis only included baseline data, we did not need to take account of attrition.

Multiple imputation for missing outcome data for smoking in late pregnancy (34–36 weeks' gestation) and at 3 months after delivery was performed using Stata's mi command, based on 20 iterations. The outcomes were imputed using multivariable logistic regression models based on the following baseline variables: age, smoking status, gestation, general health, depression, previous pregnancy, smoking in previous pregnancy, smoking

urges, qualifications and ethnicity. All baseline variables were included in the analysis in a dichotomised format. The percentage of women smoking at each outcome was obtained using Rubin's rule.[40] Where necessary, an augmented regression approach was used to overcome issues relating to perfect prediction during the multiple imputation.

## RESULTS

Figure 1 summarises questionnaire response rates. Of 1101 eligible women, 966 (88%) completed the baseline survey, and 850 (77%) consented to receive the later surveys. Questionnaires were returned by 509 (59.9%)

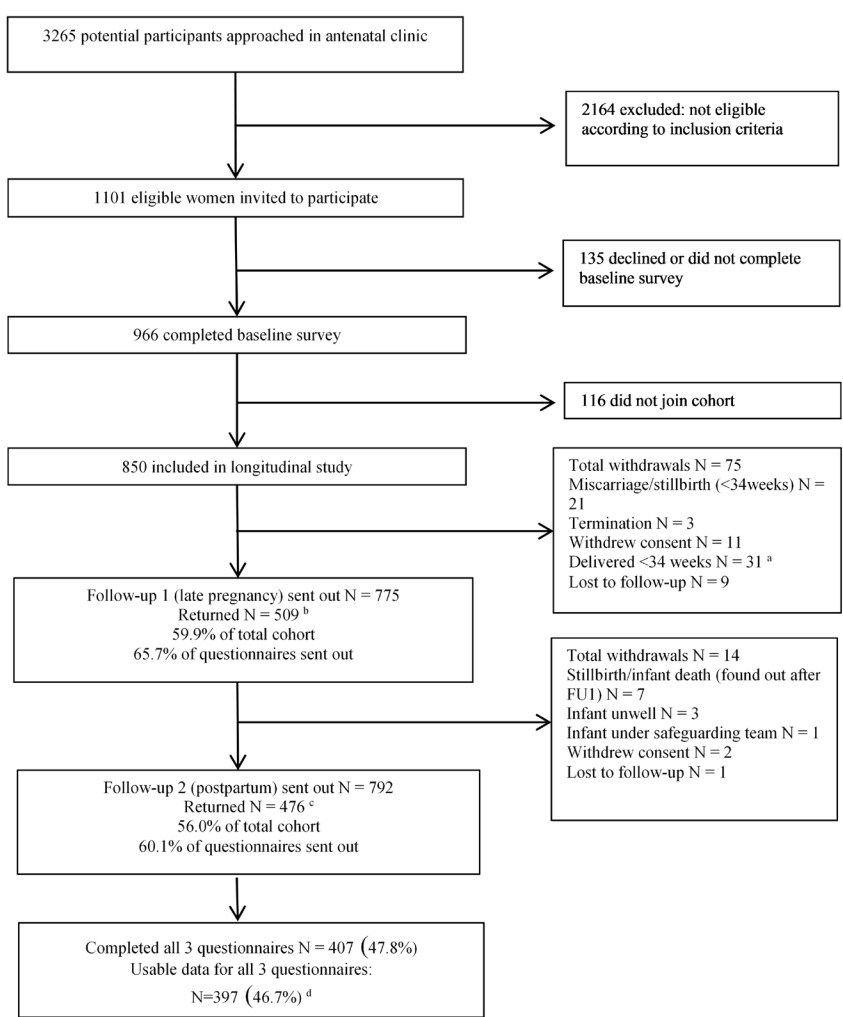

ᵃ 28 of the participants who delivered before 34 weeks gestation were sent the postpartum questionnaire

ᵇ 13 excluded who completed a follow-up 1 questionnaire, but weren't eligible (i.e. any women who was not still pregnant at 34 weeks)

ᶜ 70 participants who did not complete a late pregnancy questionnaire returned the postpartum questionnaire; of these, seven were women who delivered before 34 weeks gestation

ᵈ 10 excluded who completed all 3 questionnaires, but weren't eligible to complete follow-up 1 questionnaire (i.e. if miscarriage/stillbirth before 34 weeks gestation, terminated pregnancy or gave birth before 34 weeks)

**Figure 1** Diagram showing recruitment and flow of participants through the study.

**Table 1** Comparison of participants who completed all three questionnaires with those who completed either one or two questionnaires

| Characteristic | Completed all follow-up surveys, n=397 (46.7%) | Did not complete all surveys, n=453 (53.3%) | P value |
|---|---|---|---|
| | n (%) | n (%) | |
| Weeks' gestation (mean, SD) | 15.8 (4.1) | 15.4 (4.1) | 0.14 |
| Age, years (mean, SD) | 26.5 (5.6) | 25.2 (5.5) | <0.001*** |
| Baseline smoking status | | | |
| Current smoker | 199 (50.1) | 289 (63.8) | |
| Ex-smoker | 198 (49.9) | 164 (36.2) | <0.001*** |
| Previous pregnancy | | | |
| Not been pregnant before | 143 (36.0) | 132 (29.1) | |
| Been pregnant before | 250 (63.0) | 314 (69.3) | 0.037* |
| Partner smoking | | | |
| Partner is not a current smoker/no partner | 158 (39.8) | 187 (41.3) | |
| Partner is a current smoker | 236 (59.4) | 263 (58.2) | 0.67 |
| Current smokers only | | | |
| Reported quit attempt since learning of pregnancy/ previous questionnaire/birth of baby† | | | |
| Yes | 115 (57.8) | 157 (54.3) | |
| No | 78 (39.2) | 122 (42.2) | 0.47 |
| Heaviness of Smoking Index | | | |
| Low dependence (0–2) | 140 (70.4) | 170 (58.8) | |
| Moderate dependence (3–4) | 46 (23.1) | 100 (34.6) | |
| High dependence (5–6) | 1 (0.5) | 7 (2.4) | 0.004* |

*Significant at <0.05.
***Significant at <0.001.
†Quit attempts: at baseline, since finding out about the pregnancy; at follow-up 1, since completing previous survey; follow-up 2, since birth of baby.

in late pregnancy and by 476 (56.0%) at 3 months post-partum with 407 (47.8%) women completing all three.

Participants had similar sociodemographic characteristics to those in previous pregnancy cohorts and have been reported elsewhere.[24] Just over half (488, 57.4%) were current smokers and 729 (85.7%) of the 850 women in the cohort reported their longer-term quitting intentions (data missing for 121 (14.2%)). Of these 729 women, 424 (58.2%) planned to stop smoking permanently, 21 (2.9%) intended to stop until their baby was born and 181 (24.8%) were unsure; however, 103 (14.1%) did not plan to stop. Among the 272 smokers who reported a quit attempt at baseline, 14 (7.6%) reported stopping for >30 days, 32 (12.2%) for 7–30 days, 126 (48.1%) for 1–6 days and 84 (32.1%) for <24 hours.

Responding to all three surveys was associated with being older, less cigarette dependant, primiparous, in a planned pregnancy and being a 'recent ex-smoker' at the outset of the study (table 1).

Figure 2 shows a preliminary descriptive analysis of smoking behaviour across pregnancy within the 397 participants who returned all three questionnaires and illustrates variability in individual's smoking behaviour. Of note, 13.5% (5/37) of women who had stopped smoking in the 3 months before pregnancy were smoking again 3 months after childbirth, whereas 34.2% (55/161) of women who reported that they had quit after finding out they were pregnant had returned to smoking 3 months postpartum. As these data are not adjusted for non-response at follow-up, they may not be consistent with adjusted figures reported below.

Table 2 shows findings from univariable and multivariable analyses that investigated factors associated with baseline current smokers having reported making a quit attempt earlier in pregnancy. As these analyses only used baseline data, adjustment for attrition was not needed. After the multivariable modelling, four factors were independently associated with reporting previous quit attempts at baseline: smoking fewer daily cigarettes, agreeing that smoking during pregnancy can seriously harm the baby, being primiparous and having a planned pregnancy.

At baseline, smokers reported making a median (IQR) of 2 (1–3) quit attempts lasting at least 24 hours since

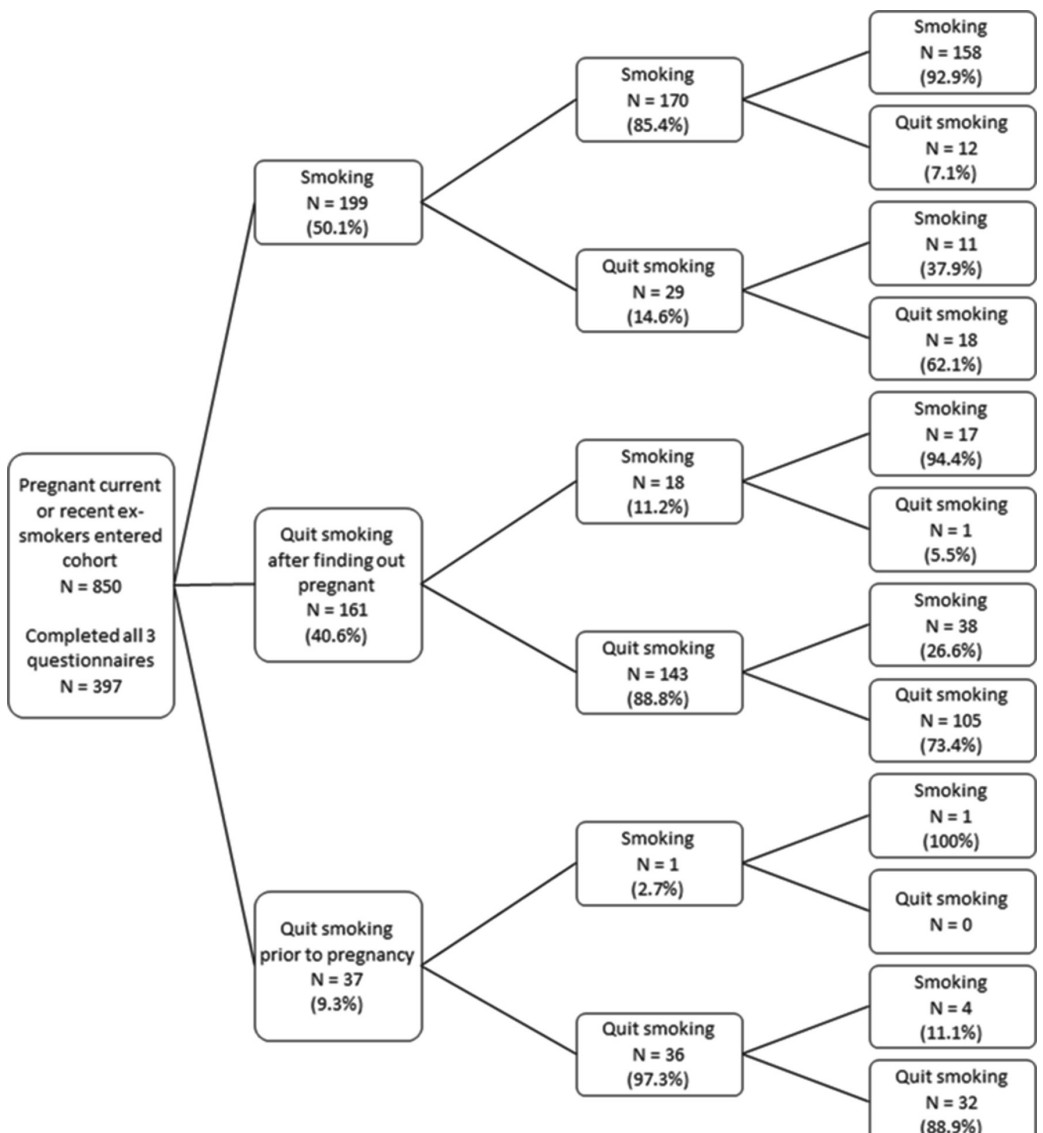

**Figure 2** Change in smoking behaviour between early pregnancy, late pregnancy and postpartum among respondents to all three questionnaires (n=397).

discovering they were pregnant; in later pregnancy, 2 (1–5) quit attempts were reported since completing the first questionnaire, and in the postpartum, 2 (1–4) quit attempts were reported since childbirth (unadjusted data). The median number of quit attempts made by those who smoked across their pregnancy (smokers who completed both baseline and late pregnancy questionnaires, n=177) was 3 (IQR 1–6); these data were highly skewed with a range of 0–60 24 hours quit attempts reported.

Table 3 shows data on smoking rates, quitting behaviour and quit intentions at the three time points adjusted for non-response, as appropriate, using multiple imputation; raw (unadjusted) data are included for reference in (supplementary appendix table s1). Adjusted figures show no evidence that smoking rates changed in pregnancy; the proportion of smokers was 57.4% (95% CI 54.1 to 60.7) at baseline and 59.1% (95% CI 54.9 to 63.4) in late pregnancy. However, by 3 months postnatally, the adjusted

proportion of current smokers was 67.1% (95% CI 62.7 to 71.5). Over half (55.7%, 95% CI 51.3 to 60.1) of smokers reported making quit attempts since becoming pregnant and 51.3% (95% CI 44.7 to 58.0) did so between early and late pregnancy; however, only 27.4% (95% CI 21.7 to 33.2) reported trying to stop after childbirth. The proportion of women who intended to try quitting within the next month fell as pregnancy progressed from 40.4% (95% CI 36.1 to 44.8) at baseline to 29.7% (95% CI 23.8 to 35.6) in late pregnancy and only 14.2% (95% CI 10.0 to 18.3) postpartum.

Adjusted data show some differences in the rates of women restarting smoking according to when they report that they initially quit. Among women who had either not smoked in the 3 months before pregnancy or during early pregnancy (before completing the baseline questionnaire), 10.2% (95% CI 6.5 to 13.9) reported smoking on the late pregnancy questionnaire and 31.2% (95% CI 25.2 to 37.2) did so at 3 months postpartum.

**Table 2** Univariableand multivariable associations with previous quit attempts in pregnancy at baseline

| Variable | Current smokers in each response category, n | Women who made a quit attempt, n (row %) | OR (95% CIs) | P value | OR (95% CIs) | P value |
|---|---|---|---|---|---|---|
| | | | **Univariable model** | | **Multivariable model** | |
| Age, years | | | | | | |
| <20 | 97 | 64 (66.0) | 1.00 | | | |
| 21–25 | 179 | 101 (56.4) | 0.63 (0.37 to 1.07) | | | |
| 26–30 | 123 | 57 (46.3) | 0.42 (0.24 to 0.74) | 0.026 | | |
| Over 31 | 86 | 48 (55.8) | 0.64 (0.35 to 1.19) | | | |
| Age (years) mean (SD) | 25.3 (5.4) | 24.9 (5.7) | 0.97 (0.94 to 1.01) | 0.11 | | |
| General health | | | | | | |
| Excellent | 68 | 45 (66.2) | 1.00 | | | |
| Good | 348 | 178 (51.2) | 0.49 (0.28 to 0.87) | | | |
| Fair | 68 | 47 (69.1) | 0.95 (0.46 to 1.97) | 0.016 | | |
| Poor | 2 | 1 (50.0) | 0.44 (0.03 to 7.5) | | | |
| Qualifications held | | | | | | |
| None | 121 | 50 (41.3) | 1.00 | | | |
| GCSEs or above | 367 | 222 (60.5) | 2.27 (1.49 to 3.46) | 0.0001 | | |
| Previous pregnancy | | | | | | |
| Yes | 346 | 169 (48.8) | 1.00 | | 1.00 | |
| No | 137 | 101 (73.7) | 3.17 (2.02 to 4.98) | <0.0001 | 2.20 (1.33 to 3.66) | 0.0019 |
| No of cigarettes smoked per day | | | | | | |
| ≤5 | 191 | 136 (71.2) | 1.00 | | 1.00 | |
| 6–10 | 151 | 86 (57.0) | 0.56 (0.36 to 0.88) | <0.0001 | 0.65 (0.39 to 1.07) | |
| ≥11 | 131 | 45 (34.4) | 0.22 (0.14 to 0.36) | | 0.28 (0.16 to 0.48) | <0.0001 |
| HSI | | | | | | |
| Low dependence | 310 | 196 (63.2) | 1.00 | | | |
| Moderate dependence | 146 | 61 (41.8) | 0.43 (0.29 to 0.64) | <0.0001 | | |
| High dependence | 8 | 2 (25.0) | 0.19 (0.04 to 0.98) | | | |
| Urge to smoke in last 24 hours | | | | | | |
| No urges | 23 | 14 (60.9) | 1.00 | | | |
| Urges | 447 | 251 (56.2) | 0.97 (0.42 to 2.24) | 0.95 | | |
| Strength of urges to smoke in last 24 hours | | | | | | |
| No urges | 31 | 16 (51.6) | 1.00 | | | |
| Weak | 334 | 194 (58.1) | 1.28 (0.60 to 2.70) | 0.82 | | |
| Strong | 103 | 59 (57.3) | 1.26 (0.55 to 2.86) | | | |
| Partner smoking status | | | | | | |
| Non-smoking partner | 111 | 57 (51.4) | 1.00 | | | |
| Smoking partner | 334 | 189 (56.6) | 0.88 (0.57 to 1.37) | 0.50 | | |
| No partner | 41 | 26 (63.4) | 1.38 (0.69 to 2.73) | | | |
| Home ownership | | | | | | |
| Rent/other | 427 | 234 (54.8) | 1.00 | 0.47 | | |
| Own home | 57 | 35 (61.4) | 1.23 (0.70 to 2.17) | | | |

Continued

**Table 2** Continued

| Variable | Current smokers in each response category, n | Women who made a quit attempt, n (row %) | OR (95% CIs) | P value | OR (95% CIs) | P value |
|---|---|---|---|---|---|---|
| | | **Univariable model** | | | **Multivariable model** | |
| **Current employment** | | | | | | |
| Not in current paid work | 324 | 163 (50.3) | 1.00 | 0.0005 | | |
| In current paid work | 164 | 109 (66.5) | 2.01 (1.35 to 2.99) | | | |
| **Usual occupation** | | | | | | |
| Manual/not applicable | 351 | 194 (55.3) | 1.00 | 0.13 | | |
| Non-manual | 75 | 46 (61.3) | 1.50 (0.88 to 2.57) | | | |
| **Ethnicity** | | | | | | |
| White British | 447 | 250 (55.9) | 1.00 | | | |
| Other | 39 | 21 (53.9) | 1.05 (0.53 to 2.10) | 0.88 | | |
| **Timing of pregnancy** | | | | | | |
| Planned | 171 | 110 (64.3) | 1.00 | | 1.00 | |
| Surprise | 312 | 158 (50.6) | 0.59 (0.40 to 0.87) | 0.007 | 0.53 (0.34 to 0.82) | 0.0045 |
| **Felt depressed or hopeless in last month** | | | | | | |
| Yes | 144 | 88 (61.1) | 1.00 | 0.21 | | |
| No | 338 | 181 (53.6) | 0.77 (0.52 to 1.16) | | | |
| **Long-term disability or mental illness** | | | | | | |
| Yes | 66 | 34 (51.5) | 1.00 | | | |
| No | 416 | 234 (56.3) | 1.25 (0.74 to 2.11) | 0.41 | | |
| **Smoking during pregnancy can harm your baby** | | | | | | |
| Disagree | 211 | 81 (38.4) | 1.00 | | 1.00 | |
| Agree | 266 | 183 (68.8) | 4.08 (2.76 to 6.02) | <0.0001 | 4.23 (2.76 to 6.48) | <0.0001 |
| **Self-efficacy in quitting** | | | | | | |
| Low | 412 | 220 (53.4) | 1.00 | | | |
| High | 47 | 35 (74.5) | 3.72 (1.68 to 8.21) | 0.001 | | |

GCSE, general certificate of secondary education; HSI, heaviness of smoking index.

However, if this information is broken down further, for those who said they quit *prior* to becoming pregnant, only 2.5% reported smoking by late pregnancy and 17.8% (95% CI 6.1 to 29.5) reported smoking 3 months postpartum. Whereas, of those who reported quitting *after* finding out they were pregnant (but before completing the baseline questionnaire), 11.6% (95% CI 7.3 to 15.9) were smoking by late pregnancy and 34.4% (95% CI 27.6 to 41.2) were smoking 3 months postpartum. By comparison, among smokers at baseline who reported not smoking in late pregnancy, 42.9% (95% CI 24.6 to 61.3) were smoking 3 months after delivery. Overall, of women who reported abstinence on the late pregnancy questionnaire, 26.2% (95% CI 20.3 to 32.2) had relapsed by 3 months post delivery.

## DISCUSSION

To our knowledge, this is the first study to use prospectively collected, longitudinal data to quantify changes in smoking behaviour through the examination of multiple quit attempts and women's intention to quit during pregnancy and postnatally. Despite over 50% of smokers reporting quit attempts across all three trimesters, there was no evidence that overall smoking rates changed between joining the study at around 8–24 weeks' gestation and late pregnancy. In smokers, intention to quit within the next month fell as the pregnancy progressed, and then fell further postpartum. Within 3 months of giving birth, around one third of women who achieved abstinence before or during early pregnancy had returned to smoking. However, we observed a trend, not

**Table 3** Smoking behaviours reported in pregnancy and the postpartum adjusted for non-response at late pregnancy and 3 months postpartum using multiple imputation

| Characteristic | Baseline (early pregnancy) | | | Late pregnancy | | Postpartum | |
|---|---|---|---|---|---|---|---|
| | n | % | 95% CI | % | 95% CI | % | 95% CI |
| Current smokers* | 488 | 57.4 | 54.1 to 60.7 | 59.1 | 54.9 to 63.4 | 67.1 | 62.7 to 71.5 |
| Reported quit attempt since learning of pregnancy/previous questionnaire/birth of baby† | | | | | | | |
| Yes | 272 | 55.7 | 51.3 to 60.1 | 51.3 | 44.7 to 58.0 | 27.4 | 21.7 to 33.2 |
| No | 200 | 41.0 | 36.7 to 45.4 | 48.7 | 42.0 to 55.3 | 72.6 | 66.8 to 78.3 |
| If have made a quit attempt, attempt lasted at least 24 hours | | | | | | | |
| Yes | 178 | 65.4 | 59.6 to 71.0 | 78.9 | 71.6 to 86.2 | 67.8 | 55.8 to 79.8 |
| No | 90 | 33.1 | 27.7 to 38.9 | 21.1 | 13.8 to 28.4 | 32.2 | 20.2 to 44.2 |
| Cigarettes per day | | | | | | | |
| 0–10 | 342 | 70.1 | 65.9 to 74.0 | 68.3 | 63.2 to 73.3 | 60.0 | 54.3 to 65.7 |
| ≥11 | 131 | 26.8 | 23.1 to 31.0 | 31.6 | 26.6 to 36.7 | 40.0 | 34.3 to 45.7 |
| How soon after waking smoke first cigarette, min | | | | | | | |
| ≤30 | 260 | 53.3 | 48.8 to 57.7 | 48.6 | 43.1 to 54.2 | 48.2 | 41.8 to 54.6 |
| ≥31 | 206 | 42.2 | 37.9 to 46.7 | 51.4 | 45.8 to 56.9 | 51.8 | 45.4 to 58.2 |
| Heaviness of Smoking Index | | | | | | | |
| Low dependence (0–2) | 310 | 63.5 | 59.1 to 67.7 | 68.1 | 62.5 to 73.8 | 65.7 | 59.1 to 72.3 |
| Moderate-to-high dependence (3–6) | 154 | 32.6 | 27.6 to 35.8 | 31.9 | 26.2 to 37.5 | 34.3 | 27.7 to 40.9 |
| Intention to quit smoking | | | | | | | |
| Intending to quit within next 30 days | 197 | 40.4 | 36.1 to 44.8 | 29.7 | 23.8 to 35.6 | 14.2 | 10.0 to 18.3 |
| Intending to quit within next 3 months/not seriously planning to quit | 252 | 51.6 | 47.2 to 56.1 | 70.3 | 64.4 to 76.2 | 85.8 | 81.7 to 90.0 |
| Urges to smoke | | | | | | | |
| How often felt urges to smoke in previous 24 hours | | | | | | | |
| No/few urges/don't know | 707 | 83.2 | 80.5 to 85.5 | 82.0 | 78.7 to 85.3 | 78.7 | 74.8 to 82.6 |
| Frequent urges (a lot of the time-all of the time) | 135 | 15.9 | 13.6 to 18.5 | 18.0 | 14.7 to 21.3 | 21.3 | 17.4 to 25.2 |
| Strength of urges to smoke in previous 24 hours | | | | | | | |
| No urges/slight-to-moderate urges/don't know | 738 | 86.8 | 84.4 to 88.9 | 82.8 | 79.0 to 86.5 | 83.0 | 79.0 to 87.0 |
| Strong-to-extremely strong urges | 109 | 12.8 | 10.7 to 15.2 | 17.2 | 13.5 to 21.0 | 17.0 | 13.0 to 21.0 |

*Includes those who report smoking regularly and those who smoke occasionally (not every day).
†Quit attempts: at baseline, since finding out about the pregnancy; at follow-up 1, since completing previous survey; follow-up 2, since birth of baby.

previously reported in longitudinal data, whereby those who quit before pregnancy may be less likely to return to smoking postpartum than those that quit on learning of their pregnancy; those that only achieved abstinence in late pregnancy appeared to be most likely to return to smoking postpartum. Women's motivation to try quitting was lowest in the first 3 months following childbirth; only around a quarter tried quitting during this time and far fewer reported intending to quit in the immediate future than had done so at either pregnancy time point.

The originality of this study is a key strength. As previously mentioned, we could find relatively few observational studies in which pregnant women's smoking behaviours were longitudinally recorded at more than

one time point in pregnancy.[12 13 15 16 19–23] Only two of these longitudinal studies followed women up postpartum,[12 13] only four reported any data on fluctuations or trajectories in smoking status[12 13 15 16] and none evaluated multiple quit attempts, often assessing only heaviness of smoking or successful versus unsuccessful quitting. All other studies investigating individuals' changes in smoking behaviour in pregnancy have asked about this retrospectively after pregnancy or at only one time point, and in contrast, we collected longitudinal data during and after pregnancy. We particularly focused on quit attempts and quitting intentions, rather than purely on smoking status at different time points, and are not aware of any other longitudinal studies that have attempted this. In addition, for the first time, we have reported 'attrition-adjusted' rates of smoking or quitting in later pregnancy and postpartum. If we had only used cross-sectional data, we might have underestimated the proportion of smokers in later pregnancy and in the postpartum. We believe that we present the most robust available data documenting changes in smoking status and quitting behaviours across pregnancy and into postpartum, for women who are not participating in an intervention study.

A limitation is that, although we followed women longitudinally, at each data collection point, we relied on self-reported data and recall of smoking behaviour in the immediate past, so we cannot be completely sure that reports are valid. However, a number of factors should have minimised any misreporting of smoking behaviour: no intervention was tested and there was no expectation that participants should try stopping; researchers emphasised that responses were of interest irrespective of smoking status and, as women completed questionnaires at three stages, they did not have to recollect their behaviour over long periods. Additionally, studies looking at both self-report and biochemically validated smoking data suggest that self-reported smoking can be both accurate and reliable.[16 41] It is possible that pregnant women who were concerned about the stigma of smoking may have avoided participation; we do not know how this might have affected findings, but women who consented to join the cohort had similar characteristics to those who declined.[24] As the survey was conducted in just two Nottingham hospitals, it is hard to say how far findings can be generalised. To help assess generalisability, we included survey items that permitted comparison with previous studies; we found our participants who continued to smoke in early pregnancy were similar to pregnant smokers enrolled in other major UK cohorts.[24] This suggests that the principal findings may apply to pregnant smokers in the UK generally. Likewise, although absolute smoking rates and smoking cessation advice and treatment may vary, pregnant smokers from other high-income countries generally have similar characteristics to those in the UK.[28 42 43] Therefore, it could be considered reasonable to extrapolate many of our findings to pregnant smokers in high-income countries generally. Although we had very high rates of eligible women joining the cohort, a further limitation

was that attrition was relatively high, with response rates to the two later questionnaires of 60% and 56%. This is a common problem with longitudinal studies,[44] and as young, pregnant smokers were likely to be a particularly difficult group to maintain contact with, we used a number of recommended methods to try to maximise response rates.[24 44 45] However, rather than simply relying on incomplete data, we have tried to address non-response bias by adjusting later surveys' findings using multiple imputation. In addition, differences in characteristics between the whole cohort and those that responded to all three surveys (table 1) need to be considered when viewing the unadjusted smoking 'trajectory' analysis shown in figure 2. Finally, we assessed smoking status at 3 months postpartum, and it is likely that some women who were abstinent at this point will have returned to smoking after this.[17 18]

The finding that most women in our cohort had quit in the early stages of pregnancy (before joining the study) and that smoking rates did not change between the second trimester and 36 weeks' gestation is consistent with cross-sectional estimates for smoking prevalence obtained in a large US study, which reported these by month of pregnancy.[46] In that study, smoking prevalence at 1 month of gestation was 26%, then between the fourth and eighth month of gestation, smoking rates for each month were 13%–14%.[46] Other retrospective studies have found that most women who successfully quit are likely to achieve this soon after finding out they are pregnant,[5 47] often within the first few days.[47] Many quit spontaneously after discovering they are pregnant.[48] Therefore, it seems that, after the early stages of pregnancy, despite still reporting quit attempts, women's smoking behaviour actually undergoes very little change.

One study found that 70% of pregnant women making their first quit attempt did so in their first trimester; however, these data were collected up to 5 years after delivery and only considered first quit attempts.[15] We found that some women made multiple quit attempts throughout pregnancy and we have previously reported that, at baseline, most reported cutting down or only smoking occasionally since becoming pregnant with less than 8% of our cohort saying that they smoked the same or more than before pregnancy.[24] Although self-reported, this reinforces findings from qualitative studies, which indicate that many persistent smokers report deliberate, and sometimes detailed, plans to cut down in their pregnancy, seeing this as a positive step and often as a route to quitting.[49] Far fewer women reported making quit attempts in the 3 months after childbirth than they did during pregnancy. Even in early pregnancy, around half of women had no intention to quit within the next 30 days; intention to quit in the short term was even lower in late pregnancy and was lowest of all postpartum. This diminishing intention to quit has not been reported before and could be considered when designing and delivering cessation interventions; for example, earlier intervention may be more successful.

We found that women who were primiparous, smoked fewer cigarettes per day, had a planned pregnancy and believed smoking during pregnancy could seriously harm their baby were more likely to have made a quit attempt during early pregnancy. These findings are comparable to previous literature examining the characteristics of pregnant smokers who successfully achieve cessation. Primiparous women have previously been found to be more likely to successfully quit smoking.[28][29][31][32] This may be because women who have smoked throughout a previous pregnancy without experiencing complications may view the risks of smoking during pregnancy differently to primiparous women, and therefore be less motivated to make a quit attempt.[29] Similarly, previous studies have found that heavier smoking is negatively associated with successfully quitting in pregnancy,[29][31][33] and heavier smokers are less likely to have high motivation to quit during pregnancy.[33] Women whose pregnancies are unintended have previously been found to be more likely to continue smoking during pregnancy,[28][35–37] and likewise, pregnant smokers who do not report concern about the effect smoking might have on the health of their unborn baby were more likely to have low motivation to quit smoking.[33] These findings identify women who are most likely to make a quit attempt and will potentially benefit the most from National Health Service support. Heavier smokers and women in second or later and unplanned pregnancies who are less likely to try quitting may require different, more intensive or tailored forms of support.

### Implications for practice

Although our data suggest that motivation to quit may be strongest in early pregnancy, some women will be receptive to quitting at any time, as indicated by their multiple quit attempts throughout pregnancy, and this confirms that it is important to discuss smoking with women at every appointment and to refer them for stop smoking support.[9] One rather surprising finding was that, in early pregnancy to mid-pregnancy, 44% (211/477) of smokers disagreed that smoking in pregnancy can harm their baby; as those who agreed with this statement were more likely to have made previous quit attempts at baseline, additional education on this issue should be considered by health professionals. Preventing resumption of smoking after pregnancy is a critical public health issue; if women restart, their lifelong health is at risk and their infants are more likely to be exposed to secondhand smoke[50] and to eventually become smokers.[51] Women often need help to resist returning to smoking after childbirth, but there are currently few effective interventions for this.[52] Women appear to be more inclined to consider quitting during pregnancy than in postpartum, and this is important when designing interventions. A potential reason for restarting smoking and for making fewer quit attempts postpartum may be that women perceive that harm to the baby from smoking is much higher during pregnancy compared with after delivery. However, some postpartum women do make quit attempts or may be planning to quit in the medium term, so engaging with them again after birth, to think about planning for this in the medium term, rather than immediately, might be a successful option. Previous studies have shown that women who quit spontaneously early in pregnancy are likely to be different and more successful than those who quit later,[48] and we found that women appear to be more likely to return to smoking after childbirth the later in pregnancy they quit. Therefore, exploring potential reasons for this, for example, demographic factors or women's intentions, could help to identify if different women may benefit from alternative approaches to help prevent relapse, perhaps by developing more tailored interventions. Although quit attempts might suggest receptivity to quitting, what is not well understood is how interest in smoking cessation support may change during pregnancy.

## CONCLUSIONS

Many pregnant women who smoke attempt quitting throughout their pregnancy, but this makes little difference to overall smoking rates. After giving birth, most smokers seem less inclined to make further quit attempts and many who quit in early pregnancy return to smoking. Women who quit in late pregnancy may be most likely to return to smoking after childbirth, while those who stopped prior to pregnancy may be least likely to relapse. It is therefore imperative to discuss smoking with women, including recent ex-smokers, throughout pregnancy and postpartum, and to continue to offer and provide specialist stop smoking support.

**Acknowledgements** The authors would like to thank all participants and staff involved in this study and Nottingham University Hospitals NHS Trust for facilitating this research. We also thank James Brimicombe for developing the study database and Rachel Whitemore for her invaluable help with this study.

**Contributors** SC helped to conceive the study, made a substantial contribution to the development of the protocol and questionnaires, assisted with day-to-day troubleshooting during the data collection phase and drafted and revised this manuscript. SO helped to design the data collection process, recruited participants into the cohort, managed the day-to-day running, assisted with data analysis and contributed to the drafting of this manuscript. JL-B contributed to the development of the study protocol and questionnaires, advised on analysis and contributed to the preparation of this manuscript. EB undertook the analyses and interpretation of data in table 2 as part of a Bachelor of Medical Sciences project and contributed to the preparation of this manuscript. LV assisted with data analysis and interpretation and contributed to the preparation of this manuscript. KB helped to design the data collection process, recruited participants into the cohort, managed the day-to-day running and contributed to the preparation of this manuscript. FN, MU, KEP and SS all contributed to the development of the study protocol and questionnaires, contributing expertise from their own particular knowledge base, and to the preparation of this manuscript. TC conceived the study and made substantial contributions to the development of the study protocol and questionnaires and to the preparation of this manuscript. All authors read and approved the final manuscript. Rachel Whitemore assisted with study administration, telephone follow-ups and data entry.

**Funding** This article presents independent research funded by the National Institute for Health Research (NIHR) under its Programme Grants for Applied Research Programme (grant number RP-PG 0109-10020). The views expressed in this paper are those of the authors and not necessarily those of the National Health Service, the NIHR or the Department of Health. SC, SO, JL-B, KB, FN, MU, SS and TC are members of the UK Centre for Tobacco and Alcohol Studies (UKCTAS), a UK Clinical Research Collaboration Public Health Research Centre of Excellence.

UKCTAS receives core funding from the British Heart Foundation, Cancer Research UK, Economic and Social Research Council, Medical Research Council and the Department of Health under the auspices of the UK Clinical Research Collaboration. SC, SO, KB, SS and TC are members of the NIHR School for Primary Care Research. TC acknowledges the support of the NIHR Collaboration for Leadership in Applied Health Research.

**Competing interests** KEP is a trustee of The Equality Trust (a registered charity) and receives occasional honoraria, all of which are donated to The Equality Trust or for student support at the University of York. In the last 5 years, TC has been paid honoraria on two occasions for speaking at meetings or conferences organised by Pierre Fabre Laboratories (a nicotine replacement therapy manufacturer).

**Ethics approval** Derbyshire Research Ethics Proportionate Review Sub-Committee gave ethical approval.

**Provenance and peer review** Not commissioned; externally peer reviewed.

**Data sharing statement** The dataset is still subject to further analyses. Relevant anonymised data available from the authors on reasonable request.

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
