## [Reviewer comments · BMJ Open]

ARTICLE DETAILS

TITLE (PROVISIONAL)	Smoking and quit attempts during pregnancy and postpartum: a longitudinal UK cohort
AUTHORS	Cooper, Sue; Orton, Sophie; Leonardi-Bee, Jo; Brotherton, Emma; Vanderbloemen, Laura; Bowker, Katharine; Naughton, Felix; Ussher, Michael; Pickett, Kate; Sutton, Stephen; Coleman, Tim

VERSION 1 REVIEW

REVIEWER	Suena H. Massey, MD Associate Professor in Psychiatry & Behavioral Sciences Northwestern University Feinberg School of Medicine United States
REVIEW RETURNED	04-Aug-2017

GENERAL COMMENTS	Summary and Recommendation This clearly-written well-powered survey study addresses a critical gap in literature– the nature and timing of quit attempts made by pregnant women. In light of the prevalence and long term risks to both mothers and children associated with smoking during pregnancy (obstetric, neonatal, neurodevelopmental, cardiovascular and metabolic), combined with the modest impact that interventions to date have had on smokers who have not quit on their own, these findings are of great relevance for prevention and public health. I believe this article should definitely be published after the following fairly minor revisions are made. 1. Introduction, Page 5 line 39: “Three prospective, longitudinal studies have reported smoking patterns in pregnancy;^{10 11 15} however, two are over 20 years old^{10 15} (including the only one to have been conducted in the UK¹⁰), the third had a smoker response rate of only 25%¹¹ and none of these studies report when in pregnancy women have tried to quit.^{10 11 15} If ‘patterns’ is construed to mean quit versus continue smoking, then this statement is not accurate – cites below contain some (not many) longitudinal studies of smoking patterns during pregnancy. The 2012 review article below includes 4 longitudinal studies – Appleton & Pharoah 1998, Maxson et al., 2011, Wakschlag et al., 2003, and Kodl & Wakschlag 2004 - 2 of which contain prospective biomarkers (Wakschlag studies). Below are 3 additional more recent longitudinal studies, one of which is specifically focused on quitting (Massey et al., 2015 in Addictive Behaviors).
--

Massey, S. H., & Compton, M. T. (2012). Psychological differences between smokers who spontaneously quit during pregnancy and those who do not: a review of observational studies and directions for future research. *Nicotine & Tobacco Research*, 15(2), 307-319.

Massey, S. H., Bublitz, M. H., Magee, S. R., Salisbury, A., Niaura, R. S., Wakschlag, L. S., & Stroud, L. R. (2015). Maternal–fetal attachment differentiates patterns of prenatal smoking and exposure. *Addictive behaviors*, 45, 51-56.

Massey, S. H., Estabrook, R., O'Brien, T. C., Pine, D. S., Burns, J. L., Jacob, S., . . . Wakschlag, L. S. (2015). Preliminary evidence for the interaction of the oxytocin receptor gene (*oxtr*) and face processing in differentiating prenatal smoking patterns. *Neuroscience letters*, 584, 259-264.

Estabrook, R., Massey, S. H., Clark, C. A., Burns, J. L., Mustanski, B. S., Cook, E. H., . . . Wakschlag, L. S. (2016). Separating family-level and direct exposure effects of smoking during pregnancy on offspring externalizing symptoms: bridging the behavior genetic and behavior teratologic divide. *Behavior genetics*, 46(3), 389-402.

However, it is possible authors meant to say that only these 3 longitudinal studies have examined the timing of more than one quit attempt. If this is the case, this should be clarified. I suggest saying, relatively few studies of prenatal smoking have been longitudinal (cite) and even fewer have examined more than one quit attempt.

2. Measurements section, Page 6 lines 15 and 52:

Authors might clarify whether 'pregnancy' alluded to in lines 15 and 52 connotes estimated date of conception versus the date the pregnancy was recognized. The former might be more relevant to prenatal exposure whereas the latter is more relevant to 'quitting because of pregnancy.'

Discussion

3. I suggest highlighting the strengths of this study more (!) and putting it in the context of other longitudinal studies that have assessed only successful versus unsuccessful quitting, not multiple attempts, though have examined maternal factors not examined here.

This study's greatest contribution is study is its longitudinal assessment of multiple quit attempts and intention, and in particular, the finding of apparent attrition in making attempts as the pregnancy progresses. This suggests that earlier intervention may be more successful, and raises very interesting possibilities including the decreasing salience of the pregnancy/fetus on motivation to quit over time. To further deepen the discussion, authors might consider pointing out that many persistent smokers report deliberate (sometimes quite detailed) plans to cut down (see cite below).

Graham, H., Flemming, K., Fox, D., Heirs, M., & Sowden, A. (2014). Cutting down: insights from qualitative studies of smoking in pregnancy. *Health & social care in the community*, 22(3), 259-267.

4. Consider moving the strengths paragraph on page 19 line 24 up, to come before the paragraph beginning with "A limitation is that..." on page 18 line 14.

	5. Page 17 line 38 “To our knowledge, this is the first UK study to use prospectively collected, longitudinal data to quantify changes in smoking behaviour during pregnancy and postnatally.” And page 19 line 24 “As previously mentioned, we could only find three previous observational studies in which pregnant women’s smoking behaviours were longitudinally recorded at more than one time point in pregnancy and these have limitations.” See #1. Again, inclusion of the additional cites is not necessary if authors specify the term ‘patterns’ to mean examination of multiple quit attempts longitudinally (rather than categorize multiple attempters as ‘women who did not quit’ as is most commonly done). 6. Nice job on the Implications for Practice section. Consider pointing out the possibility that the ‘harm to the baby from smoking’ might be perceived as much higher during pregnancy versus following delivery – as an additional explanation for postpartum relapse. Again, really nice work – a very important contribution our field! A pleasure to review.
--	---

REVIEWER	Taneisha Scheuermann University of Kansas Medical Center, United States
REVIEW RETURNED	09-Aug-2017

GENERAL COMMENTS	Introduction The rationale for the study could be strengthened by including relevant references to bolster the points made in the second paragraph of the introduction. For example, there is evidence from retrospective studies that many women continue to smoke throughout pregnancy and that many relapse during pregnancy and postpartum. Page 5, line 53 – what factors were of interest? Was there a conceptual framework to guide the factors examined? Methods Measurements – How were current smoking and occasional smoking defined? Would you provide a citation or example item for future intentions to quit? What was the time frame used for quit attempts, e.g., quit attempts in the past month, since becoming pregnant..? Results Some indication to the selection/organization of the variables for the univariable analyses for baseline quit attempts would be useful. For Table 3, the range for cpd for current smokers includes “0 -10.” Please provide a rationale for this category that includes 0. In the introduction, the authors stated that the gap to be addressed included longitudinal data on quitting and receptivity to cessation support. Their stated that their purpose was to investigate the frequency of quit attempts and factors associated with attempts and to quantify individual-level changes in smoking behavior.
---

	Revisiting these goals, first, limiting the analysis to baseline factors associated with baseline quit attempts seems to underutilize the rich longitudinal data that was collected. The authors could examine the baseline predictors of quit attempts later in pregnancy and during postpartum. If these are similar, this could be reported in the text. Second, the authors do not report on the responses to interest in different types of smoking cessation assessed at each time point (as reported in the measures and indicated by the introduction). Discussion The discussion presents the findings clearly within the context of the existing literature. Of note, assessed smoking status at 3 months postpartum – there may be additional relapse after this time period according to previous research. I was surprised that 38% of women disagreed that smoking during pregnancy can harm their babies. This could also have practice implications. Minor edits Abstract In the conclusions, “intention to quit reduces” may be clearer if rephrased as intention to quit decreases over time. Introduction Second paragraph, “we are aware of no studies” could be rephrased as being “not aware of any”.
--	--

VERSION 1 AUTHOR RESPONSE

Reviewer: 1

Reviewer Name

Suena H. Massey, MD

Please leave your comments for the authors below

Summary and Recommendation

This clearly-written well-powered survey study addresses a critical gap in literature– the nature and timing of quit attempts made by pregnant women. In light of the prevalence and long term risks to both mothers and children associated with smoking during pregnancy (obstetric, neonatal, neurodevelopmental, cardiovascular and metabolic), combined with the modest impact that interventions to date have had on smokers who have not quit on their own, these findings are of great relevance for prevention and public health. I believe this article should definitely be published after the following fairly minor revisions are made.

Comment 1. Introduction, Page 5 line 39:

“Three prospective, longitudinal studies have reported smoking patterns in pregnancy;10 11 15 however, two are over 20 years old10 15 (including the only one to have been conducted in the UK10), the third had a smoker response rate of only 25%11 and none of these studies report when in pregnancy women have tried to quit.10 11 15

If ‘patterns’ is construed to mean quit versus continue smoking, then this statement is not accurate – cites below contain some (not many) longitudinal studies of smoking patterns during pregnancy.

The 2012 review article below includes 4 longitudinal studies – Appleton & Pharoah 1998, Maxson et al., 2011, Wakschlag et al., 2003, and Kodl & Wakschlag 2004 - 2 of which contain prospective biomarkers (Wakschlag studies). Below are 3 additional more recent longitudinal studies, one of which is specifically focused on quitting (Massey et al., 2015 in Addictive Behaviors).

Massey, S. H., & Compton, M. T. (2012). Psychological differences between smokers who spontaneously quit during pregnancy and those who do not: a review of observational studies and directions for future research. *Nicotine & Tobacco Research*, 15(2), 307-319.

Massey, S. H., Bublitz, M. H., Magee, S. R., Salisbury, A., Niaura, R. S., Wakschlag, L. S., & Stroud, L. R. (2015). Maternal–fetal attachment differentiates patterns of prenatal smoking and exposure. *Addictive behaviors*, 45, 51-56.

Massey, S. H., Estabrook, R., O'Brien, T. C., Pine, D. S., Burns, J. L., Jacob, S., . . . Wakschlag, L. S. (2015). Preliminary evidence for the interaction of the oxytocin receptor gene (*oxtr*) and face processing in differentiating prenatal smoking patterns. *Neuroscience letters*, 584, 259-264.

Estabrook, R., Massey, S. H., Clark, C. A., Burns, J. L., Mustanski, B. S., Cook, E. H., . . . Wakschlag, L. S. (2016). Separating family-level and direct exposure effects of smoking during pregnancy on offspring externalizing symptoms: bridging the behavior genetic and behavior teratologic divide. *Behavior genetics*, 46(3), 389-402.

However, it is possible authors meant to say that only these 3 longitudinal studies have examined the timing of more than one quit attempt. If this is the case, this should be clarified. I suggest saying, relatively few studies of prenatal smoking have been longitudinal (cite) and even fewer have examined more than one quit attempt.

Response: We thank the reviewer for these suggestions and for bringing to our attention some references we had missed. On p5 paragraph 2, we have added the suggested references where relevant (some papers use the same cohort so in these cases we have only referenced one of the papers), have clarified the text concerning other longitudinal studies, and have stated that ours is the only study that has examined the frequency and timing of more than one quit attempt. We have included the additional suggested citations as, although our study is the only one to examine timings of quit attempts, we think it is also important to put our study in the context of other longitudinal studies. We have also dropped the word 'patterns' from the title and text of the paper as this may have been misleading. To strengthen the case we have added reference numbers 10, 11 and 17-23.

Comment 2. Measurements section, Page 6 lines 15 and 52:

Authors might clarify whether 'pregnancy' alluded to in lines 15 and 52 connotes estimated date of conception versus the date the pregnancy was recognized. The former might be more relevant to prenatal exposure whereas the latter is more relevant to 'quitting because of pregnancy.'

Response: Participants were asked about their smoking in relation to 'finding out' they were pregnant, so this would be the date the pregnancy was recognised; therefore quitting from this point is likely to be because of their pregnancy. The text has been amended to clarify this, so this now reads: 'had smoked regularly in the 3 months immediately prior to finding out they were pregnant' (p6, paragraph 2), and 'during the 3 months before finding out they were pregnant' (p7, paragraph 1). (Italics only given here for clarification, not in the text)

Discussion

Comment 3. I suggest highlighting the strengths of this study more (!) and putting it in the context of other longitudinal studies that have assessed only successful versus unsuccessful quitting, not multiple attempts, though have examined maternal factors not examined here.

This study's greatest contribution is its longitudinal assessment of multiple quit attempts and intention, and in particular, the finding of apparent attrition in making attempts as the pregnancy progresses. This suggests that earlier intervention may be more successful, and raises very interesting possibilities including the decreasing salience of the pregnancy/fetus on motivation to quit over time. To further deepen the discussion, authors might consider pointing out that many persistent smokers report deliberate (sometimes quite detailed) plans to cut down (see cite below).

Graham, H., Flemming, K., Fox, D., Heirs, M., & Sowden, A. (2014). Cutting down: insights from qualitative studies of smoking in pregnancy. *Health & social care in the community*, 22(3), 259-267.

Response: Thank you for your suggestion. We have tried to highlight the strengths of our study further in the first paragraph of the discussion (p18), and in the strengths section (p19, paragraph 2), particularly to highlight its novelty in context compared with other longitudinal studies. We have also added some discussion on cutting down behaviour.

Comment 4. Consider moving the strengths paragraph on page 19 line 24 up, to come before the paragraph beginning with "A limitation is that..." on page 18 line 14.

Response: We have moved this paragraph as suggested (now p19, paragraph 2).

Comment 5. Page 17 line 38

"To our knowledge, this is the first UK study to use prospectively collected, longitudinal data to quantify changes in smoking behaviour during pregnancy and postnatally."

And page 19 line 24

"As previously mentioned, we could only find three previous observational studies in which pregnant women's smoking behaviours were longitudinally recorded at more than one time point in pregnancy and these have limitations."

See #1. Again, inclusion of the additional cites is not necessary if authors specify the term 'patterns' to mean examination of multiple quit attempts longitudinally (rather than categorize multiple attempters as 'women who did not quit' as is most commonly done).

Response: We have clarified and amended this to say "this is the first UK study to use prospectively collected, longitudinal data to quantify changes in smoking behaviour through the examination of multiple quit attempts and women's intention to quit during pregnancy and postnatally". (First paragraph of the discussion, p18 – italics only given here for clarification, not in the text)

Comment 6. Nice job on the Implications for Practice section.

Consider pointing out the possibility that the 'harm to the baby from smoking' might be perceived as much higher during pregnancy versus following delivery – as an additional explanation for postpartum relapse.

Response: Thank you for this suggestion. We have added the following sentence: "A potential reason for restarting smoking and for making fewer quit attempts postpartum may be that women perceive that harm to the baby from smoking is much higher during pregnancy compared with after delivery." (p23, paragraph 1)

Again, really nice work – a very important contribution our field! A pleasure to review.

Response: Thank you for your very kind comments and helpful suggestions.

Reviewer: 2

Reviewer Name

Taneisha Scheuermann

Please leave your comments for the authors below

Comments and Responses:

Introduction

1. The rationale for the study could be strengthened by including relevant references to bolster the points made in the second paragraph of the introduction. For example, there is evidence from retrospective studies that many women continue to smoke throughout pregnancy and that many relapse during pregnancy and postpartum.

Response: Thank you for this suggestion. We have now added a number of references and some additional text to this paragraph to strengthen the rationale for this study (p5, paragraph 2). We have added reference numbers 10, 11 and 17-23.

2. Page 5, line 53 – what factors were of interest? Was there a conceptual framework to guide the factors examined?

Response: We tried to ensure that all factors previously been found to be associated with cessation or quitting were investigated and we included these in the questionnaires.

There wasn't any one underlying theoretical or conceptual framework, but we tried to be comprehensive as outlined above.

Methods

3. Measurements – How were current smoking and occasional smoking defined? Would you provide a citation or example item for future intentions to quit? What was the time frame used for quit attempts, e.g., quit attempts in the past month, since becoming pregnant..?

Response: Thank you for this comment. Our baseline/methods paper, published in BMJOpen, included some of this information. We acknowledge that this is important information have added some additional information to this section (p7 paragraph 1), but to add all of this, such as the exact wording of questions, would make the section very long. We will therefore upload the questionnaires to be available as an appendix to the paper, and we have amended the text in the methods to refer to these.

Further information for the reviewer:

The questionnaire responses for women used to define current or occasional smoking at baseline and follow ups were:

'I smoke occasionally, but not every day' (occasional smoker)

'I smoke every day, but have cut down during pregnancy/less than when I was pregnant' (current smoker)

'I smoke every day, about the same as before my pregnancy/as when I was pregnant' (current smoker)

'I smoke every day, and I tend to smoke more than before my pregnancy/when I was pregnant' (current smoker)

Quit attempts question (varied slightly between questionnaires):

'Since (since finding out you were pregnant/completing the first questionnaire/since the birth of your baby) have you tried stopping smoking?' (Yes/No). 'If yes, how many times have you managed to stop smoking completely for at least 24 hours?'

Quit intentions question (in each questionnaire):

'Are you seriously planning to quit?': 'Within the next 2 weeks, within the next 30 days, within the next 3 months, I am not seriously planning to quit.'

Results

4. Some indication to the selection/organization of the variables for the univariable analyses for baseline quit attempts would be useful.

Response: As far as we are aware, this is the first study to investigate characteristics of pregnant smokers who make a quit attempt, so we selected variables that previous studies have found to be associated with continued smoking or quitting in pregnancy. References to these studies are in the analysis section of the Methods (references 28-39) (p8, paragraph 1).

5. For Table 3, the range for cpd for current smokers includes "0 -10." Please provide a rationale for this category that includes 0.

Response: The participants who were asked to answer these questions included those who reported smoking 'occasionally, but not every day', therefore these women may smoke zero cigarettes some days. This information has been added to the measurements section (p7, paragraph 1) and to Table 3 (p17) as a footnote.

6. In the introduction, the authors stated that the gap to be addressed included longitudinal data on quitting and receptivity to cessation support. Their stated that their purpose was to investigate the frequency of quit attempts and factors associated with attempts and to quantify individual-level changes in smoking behavior.

Revisiting these goals, first, limiting the analysis to baseline factors associated with baseline quit attempts seems to underutilize the rich longitudinal data that was collected. The authors could examine the baseline predictors of quit attempts later in pregnancy and during postpartum. If these are similar, this could be reported in the text.

Response: We thank the reviewer for this suggestion, and we agree this may be interesting. However, like reviewer 1, we think that the paper's reporting of quitting longitudinally in pregnancy is novel and the main focus of the paper. Adding additional analyses may detract from this.

7. Second, the authors do not report on the responses to interest in different types of smoking cessation assessed at each time point (as reported in the measures and indicated by the introduction).

Response: Thank you for asking about this. The questionnaires included a number of items on interest in cessation support, however to include the responses in this paper would have made it too long. We have already planned to report this data in a future paper so have removed this information from the introduction and measures sections to avoid confusion.

Discussion

8. The discussion presents the findings clearly within the context of the existing literature. Of note, assessed smoking status at 3 months postpartum – there may be additional relapse after this time period according to previous research.

Response: We have added this sentence to the limitations: 'Finally, we assessed smoking status at 3 months postpartum, and it is likely that some women who were abstinent at this point will have returned to smoking after this.' to address this point (p21, paragraph 1).

9. I was surprised that 38% of women disagreed that smoking during pregnancy can harm their babies. This could also have practice implications.

Response: We agree this is surprising and may have practice implications. The 38% figure is actually for smokers who made a quit attempt. With all smokers this is even higher: 211/477 said they disagreed (i.e. 44%). As smokers who agreed with this statement (i.e. believe that smoking can harm their baby) are more likely to make a quit attempt we have added this to the implications section, as this is one of the only factors we found to be associated with making a quit attempt that can be addressed through education. We have added the following sentence: 'One rather surprising finding was that in early to mid-pregnancy 44% (211/477) of smokers disagreed that smoking in pregnancy can harm their baby; as those who agreed with this statement were more likely to have made previous quit attempts at baseline, additional education on this issue should be considered by health professionals.' (p23, paragraph 1)

Minor edits

Abstract

10. In the conclusions, "intention to quit reduces" may be clearer if rephrased as intention to quit decreases over time.

Response: We have rephrased this as suggested.

Introduction

11. Second paragraph, "we are aware of no studies" could be rephrased as being "not aware of any".

Response: We have rephrased this as suggested.

VERSION 2 REVIEW

REVIEWER	Suena H. Massey, MD Northwestern University Feinberg School of Medicine, United States
REVIEW RETURNED	19-Sep-2017

GENERAL COMMENTS	Authors have provided a highly-responsive revision that is more precise and better highlights the nuances of their important contribution to the literature. Bravo.
---

REVIEWER	Taneisha Scheuermann Assistant Professor, University of Kansas Medical School, USA
REVIEW RETURNED	22-Sep-2017

GENERAL COMMENTS	The authors have done excellent work in addressing all of the reviewers' comments. This manuscript is well-written and will make an important contribution to the literature. This study addresses a critical gap in the research on smoking among pregnant and postpartum women by examining women's quit attempts during pregnancy and the first three months post-partum. I am grateful for the opportunity to review this paper and its revision and I look forward to seeing it published.
---